# Towards Fully Integrated Portable Sensing Devices for COVID-19 and Future Global Hazards: Recent Advances, Challenges, and Prospects

**DOI:** 10.3390/mi12080915

**Published:** 2021-07-31

**Authors:** Tina Shaffaf, Saghi Forouhi, Ebrahim Ghafar-Zadeh

**Affiliations:** 1Biologically Inspired Sensors and Actuators Laboratory (BioSA), York University, Toronto, ON M3J 1P3, Canada; tshaffaf@yorku.ca (T.S.); sforouhi@yorku.ca (S.F.); 2Department of Biology, Faculty of Science, York University, Toronto, ON M3J 1P3, Canada; 3Department of Electrical Engineering and Computer Science, Lassonde School of Engineering, York University, Toronto, ON M3J 1P3, Canada

**Keywords:** CMOS technology, COVID-19, point-of-care, portable sensing devices

## Abstract

Since the onset of the coronavirus disease 2019 (COVID-19) pandemic, this fatal disease has been the leading cause of the death of more than 3.9 million people around the world. This tragedy taught us that we should be well-prepared to control the spread of such infectious diseases and prevent future hazards. As a consequence, this pandemic has drawn the attention of many researchers to the development of portable platforms with short hands-on and turnaround time suitable for batch production in urgent pandemic situations such as that of COVID-19. Two main groups of diagnostic assays have been reported for the detection of Severe Acute Respiratory Syndrome Coronavirus 2 (SARS-CoV-2) including nucleic acid-based and protein-based assays. The main focus of this paper is on the latter, which requires a shorter time duration, less skilled technicians, and faces lower contamination. Furthermore, this paper gives an overview of the complementary metal-oxide-semiconductor (CMOS) biosensors, which are potentially useful for implementing point-of-care (PoC) platforms based on such assays. CMOS technology, as a predominant technology for the fabrication of integrated circuits, is a promising candidate for the development of PoC devices by offering the advantages of reliability, accessibility, scalability, low power consumption, and distinct cost.

## 1. Introduction

The coronavirus disease 2019 (COVID-19) pandemic caused by the new coronavirus 2019 has affected 220 countries and territories around the world and has created a medical and socio-economic crisis [1]. When the Severe Acute Respiratory Syndrome Coronavirus 2 (SARS-CoV-2) particles bind and attack the target epithelial cells in the human body, they start to proliferate very fast and migrate to the lower parts of the respiratory system as well as the other parts of the body such as kidneys. Simultaneously, the infected epithelial cells start secreting chemical factors including cytokines and chemokines, which activate the immune system, resulting in even more subsequent symptoms and serious health issues [2]. The presence of the viral particles in the body results in the activation of the white blood cells (WBC) including neutrophils and lymphocytes to deactivate and clear the viruses through different activities including specific antibodies. COVID-19 disease has turned out to be mild and remained in the upper respiratory tracts in eight out of each ten COVID-19-positive cases, while the rest experience a more aggressive state of the disease caused by an intense immune response [3].

Diagnostic tests play a pivotal role in response to all unexpected outbreaks including COVID-19. Since the onset of this pandemic, the U.S. Food and Drug Administration (FDA) has been issuing emergency use authorization (EUA) for the diagnostic assays to be performed in the authorized laboratories in the outbreak situation to protect public health [4].

The pandemic situation reveals some shortcomings of laboratory-based assays such as requiring costly materials and specific instruments limiting many laboratories worldwide, even in the high-income countries, that are unable to purchase the instrument themselves. Furthermore, the time required for collecting and analyzing the sample to obtain an actionable result is sometimes so long that patients might lose their opportunity for treatment. Moreover, breaking quarantine for getting COVID-19 test at health centers can increase the risk of being infected, especially for high-risk people. Implementing point-of-care (PoC) self-assessment tools could control the spread of the viruses by reducing the time required to achieve an actionable result, enhancement of the level of social distancing, and the early identification of the disease.

An increasing number of recent reviews of the literature on this topic [5,6,7,8,9,10,11,12,13,14,15] have discussed the accessible diagnostic methods for COVID-19 detection, such as computed tomography (CT) of the chest, metatranscriptomics next-generation sequencing (mNSG), reverse transcription-polymerase chain reaction (RT-PCR), loop-mediated isothermal amplification (LAMP), the tools based on clustered regularly interspaced short palindromic repeats (CRISPR), and serological tests, as well as available biosensors, which are mostly optical and electrochemical.

For the detection of SARS-CoV-2, the diagnostic assays can be categorized into two main categories based on their target: The first group targets specific sequence(s) on virus genetic material and the second one senses either the coronavirus structural antigenic proteins or the antibodies generated by the human body in response to coronavirus infection [16]. The World Health Organization (WHO) has announced nucleic acid amplification tests (NAATs), particularly RT-PCR, as the gold standard strategy for detecting COVID-19 or validating the results [17]. This technology is highly sensitive with low cross-reactions and is recommended for SARS-CoV-2 sensing, especially in the initial phase of the disease before symptom onset [18]. However, it is associated with some drawbacks such as false-negative results, requiring trained technicians, biological safety level 2 laboratory, costly instruments, and limited consumptions and materials present in the market such as enzymes and tubes [19,20]. Due to the fast expansion of the infection, the diagnostic tests are preferred to have a short hands-on and turnaround time without the need for any sample preparation to be capable of being performed at home or in a PoC setting to accelerate reporting the results and decrease the risk of infection by avoiding the involvement of several laboratory members [21].

Such limitations have inspired researchers and manufacturers to take advantage of different strategies to cover RT-PCR shortcomings and conjugate NAAT tests with other strategies such as protein-based assays. By the day, there are many rapid SARS-CoV-2 antibody tests manufactured or in development to be performed either in hospital laboratories or near PoC. Table 1 shows FDA EUA-approved protein-based assays that have been commercialized for COVID-19 detection. These assays require a shorter time duration, less skilled technicians, and face lower contamination [19], leading us to focus on protein-based assays which can also be designed to be performed in the PoC setting.

Despite the commercialization of many PoC devices for COVID-19 detection (as mentioned in Table 1), some challenges such as high-throughput measurement, short turnaround time, high precision, reliability, and low cost of the PoC platforms are still under investigation. Moreover, the possibility of sample collection and sending the report directly from patients to centralized clinicians, especially during a time of effective social distancing measures, is still an open subject to study which can help to control and fight against such pandemics.

Standard complementary metal-oxide-semiconductor (CMOS) technology by offering the striking features of reliability, accessibility, considerably low cost, low power consumption, and most importantly scalability and the rapid design-to-product cycle is the best alternative technology to develop PoC devices during an urgent pandemic situation such as COVID-19. This technology allows for the monolithic integration of a large number of high-speed biosensors and actuators on a single chip and consequently gives the opportunity of high-throughput measurements in a short time. Moreover, reduced parasites and noises help to achieve higher signal-to-noise ratios (SNRs) and higher resolution. Considering the recent advances in the development of low-noise, high-speed, or high-frequency electrical circuits using CMOS technology and the huge investment in CMOS foundries, this cutting-edge technology is a promising candidate for the new generation of PoC platforms with the capability of wireless data transferring and the possibility of low-cost batch fabrication in urgent situations, so that they would be affordable for the end-users.

Therefore, in this paper, we put forward the potential of fully integrated CMOS-based technologies as an alternative solution to address the aforementioned challenges of the existing PoC devices. It is believed that a combination of the current technologies can potentially result in developing more promising detecting tools as a rapid, reliable, and adaptable diagnostic tool for detecting coronavirus or any other types of pathogens. Hence, unlike the other reviews, in this paper, we aimed to discuss the currently approved diagnostic tests for COVID-19 detection along with the functionalization strategies which have been employed for viral detection, either for COVID-19 or other viruses. These surface modifications could be used and implemented with the sensing devices which have not been employed for COVID-19 detection yet. By changing the capture molecules employed on the surface of the device and employing the suitable recognition elements specific for the target of interest, the device would be easily adaptable for detecting a large group of viruses.

As depicted in Figure 1, a fully integrated PoC device includes a disposable electronic biosensor (cartridge) and a handheld reader. This biosensor is incorporated in a microfluidic structure to prepare the sample, extract the target biological cells or molecules (e.g., viruses or antibodies), and direct them towards a sensing system. This system features a sensor and an interface circuit. A biorecognition element (BRE) is coated on the top of the sensor to selectively detect the target biomarker. The required custom-made integrated sensors and circuits for PoC testing devices can be developed using CMOS technology.

Recent decades have witnessed unprecedented advances in CMOS sensors using optical [26], electrochemical [27,28,29], and magnetic [30,31] techniques alike for a variety of applications such as for the detection of different viruses (such as human respiratory viruses [32,33], Zika virus [27], and dengue virus [30]), as well as monitoring various bacteria (e.g., *Streptococcus pneumoniae* [34], *bacillus globigii* [35], *Staphylococcus epidermidis* [26], and *Escherichia coli* [28]) and detecting parasites (such as *Plasmodium falciparum* malaria diagnosis [29]). There are other opportunities for reconfiguring these devices for the diagnosis of similar diseases. The electronic parts of these sensors can be used for similar applications but sensor calibration and the normalization of the design metrics such as dynamic range, resolution, the limit of detection (LoD), SNR, and alike would be different. If the current materials and methods reported for COVID-19 detection are CMOS-compatible, meaning that they can be fabricated and implemented by CMOS technology, this technology can open a new avenue to develop more efficient PoC platforms for detecting this virus or similar ones. Thus, in this paper, after reviewing the protein-based techniques and the materials reported for COVID-19 detection (which are more practical than nucleic acid-based techniques), the potential of CMOS biosensors to be adapted to this application is discussed.

The rest of the paper is organized as follows: In Section 2, the target protein biomarkers and the related methods for the detection of COVID-19 are introduced. In Section 3, the BREs and surface materials that have been reported for respiratory infections are discussed. Section 4 gives an overview of the various types of CMOS biosensors, such as optical, electrochemical, and magnetic sensors, and their circuit design strategies as well as their potential to be adapted to the protein-based assays discussed in Section 3. These sections are followed by a conclusion in Section 5.

## 2. Protein-Based Tests for COVID-19 Detection

Various technologies have been reported to develop reliable assays as PoC diagnostic tools to target specific biomarkers for COVID-19 detection. These protein biomarkers are categorized into two main groups of antigens and antibodies or so-called immunoglobins (Ig). The novel coronavirus particle (Figure 2) houses four main structural proteins of nucleocapsid (N), spike (S), membrane (M), and envelope (E), which act as potential biomarkers for COVID-19 detection. Furthermore, the measurement of the specific antibodies produced in the human body in response to viral antigens, is an alternative strategy employed to detect the viral infection. The human immune system, particularly B-cells, produce specific antibodies against viral structural proteins, particularly two immunoglobins named IgG and IgM. These protein antibodies are specific targets for the detection of both current and previous COVID-19 infections using antibody immunoassays [36].

Protein-based tests are mostly based on enzyme-linked immunosorbent assay (ELISA) and lateral flow immunoassay (LFIA) cassettes, as seen in Figure 3. ELISA is a routine laboratory-based method for antibody testing by using specific enzymes and specific capture antigens for SARS-CoV-2 biomarkers [37] while LFIAs implement anti-human antibodies immobilized on a nitrocellulose membrane to sense the presence of the targeted specific SARS-CoV-2 antibodies using label and anti-human conjugates. Specimens such as nasal and nasopharyngeal swabs are collected by inserting and rolling the swab in the nose or nasopharynx, respectively. Next, the swab including the sample is either directly processed or immediately placed in a sterile tube containing a transport medium for further analysis [38]. LFIA strips separate the biomarkers in the sample using a chromatographic system and capillary flow and detects them based on specific interactions between the capture molecule and the virus protein biomarker. The colloidal gold nanoparticles (AuNPs) are employed in this technology as reporters to visually detect the biomarker in the solution [39].

To obtain the characteristics of these tests, their obtained results are compared with a reference method. For COVID-19 detection, the reference method is the reference panel established by FDA. This panel consists of standardized material useful for determining and comparing the sensitivity and cross-reactivity of the developed assays for COVID-19 detection [40]. Table 1 compares the main protein-based technologies used in the test kits approved by FDA EUA in 2020. In this table, the columns represent the underlying strategy for developing each test kit, its specific biomarker(s), sensitivity and specificity of the device, possible cross-reactions with other viruses and the probability of a false result, duration of the test, Positive Predictive Agreement (PPA) and Positive Predictive Agreement (NPA), which are defined as below. In these definitions, A, B, C, and D are the true positive, false positive, false negative, and true negative.

Sensitivity (A/(A + C) × 100) is the probability of indicating COVID-19 among the infected casesSpecificity (D/(D + B) × 100) is defined as the fraction of people who are not infected by SARS-CoV-2 and have a negative test result.PPA(A/(A + B) × 100) is the probability of achieving a true positive result.NPA(D/(D + C) × 100) is the probability of achieving a negative positive result.Cross-reaction is defined as the reaction of a specific antigen with specific antibodies which are developed to target another antigen.LoD is the lowest number of biomarker copies that can be detected by a method.

### 2.1. Antigen Testing

Antigen immunoassays rely on targeting structural proteins in the viral particle, particularly N and S proteins, in nasopharyngeal or nasal samples by using SARS-CoV-2 specific BREs [41]. Antigenic tests are rapid and low-cost LFIAs suitable for the diagnosis of active infection for suspected people and individuals who were in contact with COVID-19 positive cases or who experience symptoms similar to infected individuals. These LFIAs either employ fluorescent reporters for visual detection or in a reader-based manner or use AuNPs-conjugated antibodies to detect the presence of the virus in the control line on the strip when the validity of the test is confirmed using the control line (Figure 3). As of April 2021, only 16 SARS-CoV-2 antigen diagnostic kits have received FDA emergency approval to be performed in a PoC setting [42].

It is noteworthy to mention that the clinical performances of the antigen detecting tools are largely dependent on various affecting factors as well as the patient’s situation. For instance, assuming a sample is collected in a disease phase when the viral load is high, the rapid antigen test will indicate its best performance. The best timing for antigen targeting is the early days after infection due to the highest viral load [43]. As demonstrated in Table 1, the antigen detecting fluorescent immunoassay targets virus N antigen with a very high sensitivity of 100% in the first days after infection. However, with the reduction of the viral load in the next stages of the disease, the sensitivity of the device is reduced and the false-negative results may be reported [25].

Although antigen immunoassays are usually sensitive and provide the results in a significantly shorter duration of time compared with the reference technologies, they suffer from moderate specificity and probably miss the active COVID-19 infections and report false-negative results [44,45].

### 2.2. Antibody Testing

To date, various antibody testing immunoassays have been developed and have become widely available for SARS-CoV-2 detection. These strategies are also more informative while performing for evaluation of previous infections, the body’s immune status, immune response, and when employed as screening tools for testing the rate of the disease’s prevalence [17,46]. COVID-19 antibody detection is mostly based on ELISA and LFIA test kits. ELISA is widely used as a high-throughput laboratory-based technology, employing two different antibodies specific to the target biomarker, a primary detection antibody and a second enzyme-labelled antibody, such as the anti-Nucleocapsid specific antibody, for COVID-19 detection. Enzymes such as alkaline phosphatase (ALP) are used for activation of a substrate material such as PNPP (p-Nitrophenyl Phosphate, Disodium Salt) in the solution and emitting signals from it [47,48]. In general, ELISA test kits demonstrate a higher sensitivity (and specificity) in comparison with LFIA test kits. However, the duration of ELISA tests is between 1h and 5h, which is significantly higher than the rapid LFIAs, which report the results in less than 20 min. LFIA tests also require very small amounts of the sample. Additionally, the requiring of sample preparation steps, manual procedures, and high workloads have limited ELISA applications, especially as a rapid PoC test [37].

To address the above challenge, other techniques have been used to decrease the test’s complexity and increase its speed. For instance, LFIAs are paper-based, simple, rapid, and cost-effective devices, and their low complexity, portability, and fast processing have made them popular to be conducted as PoC antibody testing devices [49]. As seen in Table 1, the sensitivity, specificity, PPA, and NPA of the ELISA technique are all better than LFIA tools, but the duration of the ELISA is at least six times higher than LFIAs. The kits developed using the same LFIA-AuNP technology could exhibit significant variation in their quantities as well. For instance, one of the LFIAs could achieve a PPA of 93.8% and NPA of 96.0% [22] while a similar LFIA assay displays a low PPA of 60.0% but a high NPA of 98.8% [24].

These broad variations could be due to the testing procedure, sample type, disease phase, as well as the quality of the developed kits and employed materials. Unfortunately, the LFIA cassettes still present low sensitivity in the clinical setting, with a high number of false-negative results [50,51]. One important challenge associated with the detection of COVID-19 using LFIA is the close relationship between disease stage and antibody variations. As many of the SARS-CoV-2-specific antibodies are not high enough for detection in the first week after the infection, a high rate of false negatives might be obtained during the first stages of the infection. Over time, the secretion of the antibodies is elevated, which results in the achieving of higher accuracy by applying these tests [49,52].

Antigen and antibody-based kits have widely been employed during the current pandemic. Nevertheless, due to the limitations of the current technologies and the urgent need for the development and mass production of more reliable sensing devices, electronic biosensors can be considered as alternative simple PoC tools for early disease detection in the current and future pandemics. Such accurate, portable, and cost-effective devices have the potential to be employed in combination with the current detecting technologies, either as a detection strategy or used for on-site detection of the infection by reporting accurate results to overcome the current shortcomings in the disease diagnostics.

Up until now, a considerable number of companies have received approval for their products to be commercialized for COVID-19 testing. However, some drawbacks are associated with these devices, such as high LoD and, in some cases, low specificity and sensitivity. It is expected to have more user-friendly and accurate diagnostic tools replaced with the current devices to be able to both qualitatively and quantitatively detect the infection by targeting specific antigens and antibodies at home. Additionally, there is a growing demand for cost-effective, ultrasensitive, portable, and simple tools for the early detection of viral infection, and batch production and universal distribution of these portable tests are of critical importance. Developing more promising surfaces for the protein-based detection of the viral infection helps the device to capture the target molecules more specifically and in a lower concentration, which has a significant role in the applicability of the device as an early detecting tool. In the next sections, we discuss these promising alternative techniques to develop miniaturized and fully-integrated antigen and antibody-based biosensing devices.

## 3. Biorecognition Elements

Currently, the main goal of the academic society is not only overcoming the current SARS-CoV-2 spread but also to be prepared for future hazards by developing reliable, cost-effective, adaptable, and portable biosensing tools [53]. Biosensors are devices that include a transducer to produce a signal and measure the concentration of the target biomolecules such as the whole virus particle, viral structural proteins, and nucleic acid [54,55]. To date, biosensing devices have been widely employed for detection purposes including the respiratory diseases caused by coronaviruses. Combining these tools with the current biological detection technologies can overcome the abovementioned challenges and form alternative methodologies for biomedical applications. Such devices with the merit of being high-throughput, mass-produced, rapid, and accurate are simple to use out of the laboratory or hospital settings without requiring any technical training or costly material [56,57]. Considering their advantages, the biosensors have attracted the attention of many researchers to be developed as the next generation of detecting tools for biomedical applications [58]. For this aim, protein structures, including IgA, IgG, IgM antibodies, and viral antigens act as promising target molecules for PoC tools with the ability to be captured using the novel biosensors without requiring any time-consuming and complicated manual steps, e.g., reverse transcription, PCR amplification or sequencing, compared with the current laboratory-based infection detection methods [59]. Capacitive biosensors integrated with CMOS readout systems are one of the novel interesting strategies for the highly accurate and label-free detection of specific biological structures [28].

Up until now, different miniaturized bioassays have been developed for diagnostic purposes based on different sensing mechanisms; however, the focus of the research and the market are mostly on optical and electrical biosensing devices [60]. In such techniques, the general structure of the biosensor is based on a BRE, an electrode, and another unit for processing signal (Figure 4) [61]. The sensing area of the sensors is functionalized with specific BREs to achieve the adhesion of specific target proteins or viral particles [62]. Hence, they generally diagnose the presence of the desired target when it is physically attached to the specific receptors immobilized on the surface. This fact demonstrates the significant importance of choosing a suitable biosensing element for developing the optimal biosensor [63].

Among the currently used BREs, antibodies are one of the most prominent and reliable biosensing elements for the accurate detection of the whole viral particle or their structural proteins, and sequester them from the solution [64]. The binding of these capturing antibodies to the transducer surfaces allows the production of electrical signals from chemical ones. The other important receptors are viral antigens which selectively capture the specific antibodies, which are secreted in the blood against a specific virus [65]. In the remainder of this section, we discuss the proposed biosensing tools with specific BREs for the detection of respiratory infectious viral diseases, including COVID-19.

### 3.1. Antibody

Antibodies have been one of the interesting and preferred BREs for the development of the PoC biosensing devices targeting viral antigens due to their robustness, wide range of binding ligands, specificity, and high affinity to bind to their target [66]. On the other hand, for serological detection, recombinant viral antigens are beneficial BREs due to forming a homogenous receptor layer on the surface and being easily produced [67]. Table 2 summarizes the characteristics of the recently developed immunosensors, employing either antibody or antigen recognition elements for the detection of viral infections causing respiratory diseases. The columns introduce the main biosensing technologies and their surface materials for targeting various viral antigens. Additionally, the table depicts a comparison between the employed functionalization strategies using different chemical procedures and linkers. These devices are functionalized with specific antibody molecules which bind to the viral structural protein(s) and have been demonstrated to detect various concentrations of the target antigens based on both the nature of the employed materials and the quality of the device, considering the probable errors. The overall combination of these characteristics results in a detection limit for the diagnostic devices; the better the design, the lower the LoD. Such devices are of critical importance, since, by replacing the capture molecules, they have the potential of being adapted for detecting any novel viral infections in future hazards.

Antibodies have been widely used as receptors for disease detection purposes including respiratory diseases. After the coronavirus spread, researchers have made several attempts to adapt these biosensors for COVID-19 diagnostics [57]. Seo et al. have reported one of the very promising graphene field-effect transistor (G-FET)-based biosensing devices for SARS-CoV-2 detection (Figure 5) [59]. Firstly, Poly (methyl methacrylate) (PMMA) C4 was spin-coated onto graphene sheets to produce PMMA/graphene. Subsequently, the PMMA/graphene layer was added on top of the SiO_2_/Si substrate of the FET sensor. To functionalize the surface, the fabricated graphene-based sensor was modified with PBASE and the graphene sheet was conjugated with the SARS-CoV-2 anti-S-specific antibodies. This portable, sensitive, and rapid FET sensor was capable of detecting SARS-CoV-2 with no cross-reaction with other respiratory viruses such as MERS-CoV and SARS-CoV, and the procedure did not require any sample preparation, pretreatment, or labelling complex steps. On the other hand, Zhang and colleagues developed another rapid G-FET immunosensor that could detect positive COVID-19 cases in about 2 min [68]. The surface of this biosensor was modified with antibody against SARS-CoV-2 S protein (S1 subunit) (CSAb) to capture the virus S protein antigen. The results demonstrated that the CSAb-modified G-FET devices detect the SARS-CoV-2 virus with higher sensitivity compared with the antigen-coated biosensor. In another attempt, they used a combination of ACE2 receptors (negatively charged) and CSAb (positively charged) to increase the sensitivity of the device. For this aim, they first synthesized single crystal graphene on single crystal Cu to form the G-FET. Afterward, they functionalized the surface area with either ACE2 receptor or CSAb as the BRE. By adding the COVID-19 positive samples, the S1 protein (positively charged) was attached to the graphene-surface immobilized CSAb/ACE2 receptors. This antibody–antigen reaction resulted in an alteration in conductance/resistance, which will be subsequently read out electrically.

Another proposed SARS-CoV-2-detecting FET sensor is based on single-walled carbon nanotube (SWCNT) technology [69]. In this device, the gold electrodes were covered with Si/SiO_2_ substrate and semiconducting SWCNTs were deposited between the gold electrodes. The SWCNTs were then modified with SARS-CoV-2-specific antibody using 1-ethyl-3-(3-dimethylaminopropyl) carbodiimide (EDC)/N-hydroxysulfosuccinimide (sulfo-NHS). This device had a very low LoD of 0.016 fg/mL for N protein and 0.55 fg/mL for S protein.

FET-based immunosensors have also been used for the detection of other respiratory infections previously. As an illustration, a label-free FET-based device was developed for the electrical detection of SARS-CoV N protein using antibody mimics as receptors [73]. The Si/SiO_2_ surface of the device was spotted with fibronectin probes and subsequently submerged in 6-phosphonohexanoic acid to attach the phosphonic acid residues to the surface. Subsequently, the functional groups of carboxylic acid were activated by adding EDC and the fibronectin probes were bound to the surface. This biosensor was able to detect 100 nM of the viral N proteins in the sample. Dual-channel FET-based immunosensors have also been fabricated for Influenza A virus detection. Hideshima et al. developed a glycan-immobilized sensing system that could directly target human Influenza A viral particles in the biological fluids [75]. In this study, they proposed a different functionalizing strategy, immobilizing host cell surface-mimetic glycan on the sensing surface. For this aim, the surface area was modified with sialic acid-α2,6-galactose and sialic acid-α2,3-galactose separately. Surface glycoproteins of the viruses recognize these sialic acid-terminated glycans and get caught by them after adding the positive sample to the immunosensor (Figure 6).

With one recent study, COVID-19 detection has become possible using an embedded metal–oxide–semiconductor field-effect transistor (MOSFET) [70]. To functionalize the Au-plated carbon electrodes with SARS-CoV-2 specific antibodies, Au–S bonding was first formed by adding thioglycolic acid (TGA). Afterwards, surface functionalization was formed using two subsequent chemical procedures of submerging the TGA-functionalized electrode in N,N0-dicyclohexylcarbodi-imide and N-hydroxysuccinimide, followed by adding both monoclonal and polyclonal spike antibodies. This system was designed with a printed circuit board (PCB) to collect the signals and read-out digitally. As demonstrated in Table 2, other types of electrochemical immunosensors have also been developed for detection of the respiratory infectious diseases, including Nanonet-FET [76] and SiNW-FET [77], with a various LoD for targeting the virus based on their design and surface functionalization.

Other than electrochemical biosensors, optical immunosensors have also gained the attraction of researchers for diagnostic purposes. A plasmonic fiber-optic absorbance biosensor (P-FAB) has recently been proposed for both labelled and label-free COVID-19 detection by functionalizing the fiber-optic probe in the U-bent sensing region. They preferred using the thiol-PEG-NHS combination as the suitable coupling procedure for surface functionalization [71]. Localized surface plasmon resonance (LSPR) sensors are another example of the optic biosensors which were functionalized with Anti-SARS-CoV N protein for SARS detection, demonstrating the ability to detect virus concentrations as low as ∼1 pg/mL [74].

### 3.2. Antigen

Although antibodies have been employed as the BREs in most of the currently developed biosensors, antigens are another source of protein bio-receptors capable of targeting specific targets in the biological samples (Table 3). Abdelhadi et al. have recently reported a portable optical surface plasmon resonance (SPR) for the detection of specific antibodies against SARS-CoV-2 N antigens in 15 min [78]. The modification was completed by adding NHS and EDC to the gold SPR surface and subsequent immobilization of SARS-CoV-2 nucleocapsid recombinant (rN) protein. Electrochemical biosensors have also been employed for the detection of respiratory infectious diseases.

In another research, Layqah and colleagues reported an electrochemical immunosensor for Middle East respiratory syndrome coronavirus (MERS-CoV) and human coronavirus (HCoV) detection [79]. After the deposition of the AuNPs on the electrode surface, it was treated with cysteamine and then human coronavirus (HCoV) or MERS-CoV antigens were incubated on separated electrode surfaces on a simple chip. The functionalized surface was demonstrated to be highly sensitive with very low LoDs of 0.4 pg.mL^−1^ to 1 pg.mL^−1^ (Figure 7).

As highlighted, antibody and antigen immunosensing devices have the potential to become batch produced and distributed as accurate, cost-effective, portable, and adaptable PoC and at-home devices for disease detection purposes. In general, the specific affinity between antigen and antibody results in the formation of conjugates, which produce the specific signals that can be read out and reported for the positive samples.

During the last two decades, the modification of the surface of different sensors has significantly advanced and many promising materials have been developed to be immobilized on top of the surfaces. The presence of more reliable deposition technologies and chemical materials has improved the process of commercializing advanced tools with more effective BREs, and such strategies will be used for developing new surface chemistries based on new demands. Although they have been widely beneficial for capturing the desired protein structures, surface modification strategies are associated with some challenges such as the immobilization of the capture molecules, their specificity and orientation, and also the stability of the coating layer. As an illustration, some of the current COVID-19 diagnostic devices are not able to distinguish between SARS-CoV and SARS-CoV-2 viruses due to their low specificity, including the low specificity of the immobilized capture molecules. Hence, in the future, specific and promising chemical structures need to be developed for a more accurate detection of the desired targets.

As described in this section, numerous biosensors have been designed and fabricated based on this binding affinity, which has successfully sensed the presence of either virus antigens or anti-virus antibodies in the bio-samples. Such biosensing settings are combined with specific readout systems, which are described in the next section. As explained in the next section, when coupled with sensors, these biosensing devices can act as potential biosensors, not only for COVID-19 detection, but also for fighting the probable viral spreads in the future.

## 4. CMOS Sensors and Circuits

According to the reported articles reviewed in the previous sections, such as [59,68,69,70], electrochemical and optical techniques have attracted much attention for COVID-19 applications. In this section, we discuss various CMOS integrated circuits and sensors for the diagnosis of diseases caused by different infectious agents including viruses, bacteria, and the like, which are potentially suitable candidates for the development of CMOS-based PoC devices for specific infectious agents such as SARS-CoV-2. It is noteworthy to mention that if the surface materials reported in the previous section could be used in the CMOS fabrication process, the CMOS circuits reviewed in this section have the potential to be incorporated with the PoCs for COVID-19 applications [80,81]. Hence, this section is mainly focused on the readout circuits which have the potential of being implemented with CMOS. Table 4 compares such circuits and summarizes their characteristics, including the CMOS process, the number of sensors in each array, and power consumption. Herein, CMOS-based devices are categorized into three groups, including optical, electrochemical (e.g., impedimetric, capacitive, voltammetry, amperometry, potentiometry), and magnetic techniques. Moreover, we briefly discuss the advantages of these sensors for viral detection.

### 4.1. Optical Techniques

The principle of a fluorescence-based biosensor can be seen in Figure 8a. The emitted signal is transduced to an electrical signal by a transducer such as a photodetector, which can be fabricated with an embedded PN-junction in the standard CMOS technology. To integrate the fluorescence module into a CMOS chip, it should be considered that the metal layers above the photodetector do not block the optical signal. Additionally, the excitation signal with stronger intensity than the fluorescent signal can saturate the photodetector and leads the system to malfunction. One of the most reported solutions to tackle this issue is the use of an optical filter atop the CMOS chip as shown in Figure 8a [82]. Song et al. [35] incorporated the ELISA technique with a CMOS chip featuring an array of 4×4-photodiodes. This laser-induced fluorescence (LIF) sensing device employed silica capillaries for immunosensing and successfully detected a single intact *B. globigii* spore. As seen in Table 4, they achieved an LoD of 0.55 cells/probe. In this device, enzymatic amplification following immune-complex formation helped to achieve a high sensitivity without the need for bulky optical systems. A narrow band-pass filter was used to remove the diode laser scattering. Figure 8b,c illustrate the diagram of this system and the detection of *B. globigii* in an antibody-immobilized capillary reactor utilizing it.

The smartphone CMOS sensors have recently been reported for fluorescence imaging by many researchers. Among them, Natesan et al. [83] employed a flow cell assay cartridge and a smartphone fluorescent reader for detecting antibody binding to twelve essential antigens immobilized in a microarray on a microfluidic chip, and monitoring the infections caused by Marburg and Ebola filoviruses. Zeinhom et al. [84] reported a smartphone-based fluorescence imager including a long-pass thin-film interference filter, high-quality insert lenses, and a compact laser-diode-based photo source for *E. coli* O157:H7 detection based on a sandwich ELISA.

The chemiluminescent signals emitted from chemiluminescent tags can also be detected by a photodetector. The tag averts the external light source to excite the chemical reaction. As a result, the saturation of the photodiode is prevented without an optical filter and the distance from the tag to the surface of the photodetector is shortened [82]. Baader et al. [34] developed polysaccharide microarrays and a CMOS-based electric signal readout process for the chemiluminescence-based detection of anti-polysaccharide IgG antibodies in human blood serum. In this system, unmodified pneumococcal polysaccharides from *S. pneumonia* were directly printed onto silicon photodiode surfaces (see Table 4).

In another effort to develop an optical CMOS biosensor for disease detection, Pilavaki et al. [33] designed a lateral flow immunochromatographic assay (LFIA) reader for detecting Influenza A nucleoprotein by designing a low-power CMOS image sensor with 4 × 64 pixels (see Table 4). Figure 8d illustrates the pixel and processing architecture of this reader. With uniform illumination at a wavelength of 525 nm, and 67 frames per second (fps), the total output referred noise and total power consumption of the chip was 1.9 mV_rms_ and 21 µW, respectively.

CMOS optical and image sensors have provided low power dissipation, high specificity, operational simplicity, and cost-effectiveness in comparison with image sensors using other technologies. CMOS sensors can be integrated with other processing circuits, control systems, and analog-to-digital converters (ADCs) [88] for high accuracy and high-speed detection of the target disease.

### 4.2. Electrochemical Sensors

Electrochemical techniques offer real-time and label-free measurement detection, and the simplicity and scalability of these sensors help to adapt them to integrated equipment. The interfacing of other electrochemical biosensors including impedimetric, capacitive, amperometric, voltammetric, potentiometric, and the like can be shown in a general schematic, such as Figure 9a, in which the required sensing electrodes can be easily formed on the top metal layer of CMOS technology and also downscaled to develop a multi-targeting array of electrodes [82].

#### 4.2.1. Impedimetric Sensor

Impedimetric sensors measure impedance changes following probe-target binding and biomedical reactions on their electrode surface. These sensors are useful for the real-time label-free detection of some biospecies, such viruses [89] and bacteria [90]. For example, Couniot et al. [90] used atomic-layer-deposited (ALD)-Al_2_O_3_ passivated microelectrodes and lytic enzymes for impedimetric detection of whole-cell bacteria. The same group [91] proposed an oscillator-based capacitance-to-frequency converter (CFC) integrated with on-chip Al/Al_2_O_3_ IDEs for whole bacterial cell detection (*S. epidermidis*) in high-conductive buffers. Figure 9b demonstrates the packaged chip proposed by this group. In another study, they [26] reported a 16×16-arrayed Al_2_O_3_ capacitive biosensor with on-chip IDEs and a capacitance-to-voltage converter as the readout circuit, which works based on the charge sharing principle and achieved a sensitivity of 2.2 mV/bacterial cell or 55 mV/fF. As seen in Table 4, the LoD and SNR of this sensor were seven bacteria (or 450 aF), and 37 dB, respectively.

#### 4.2.2. Capacitive Sensor

Capacitive biosensors measure the variations of dielectric properties and/or the thickness of the dielectric layer at the electrode-solution interface. These biosensors usually have the privilege of lower complexity compared to impedimetric biosensors, because they only measure the capacitive component of the impedance.

As mentioned in Table 4, Balasubramanian et al. [86] proposed a silicon-based capacitive sensor for virus detection, which employed a differential scheme of measurement, using a sense amplifier circuit to compare the capacitance values of a sensing capacitor and a reference capacitor. They used an ELISA assay to detect a bacterial virus M13KO7 and immobilized anti-M13KO7 on a silicon wafer to capture M13KO7.

Ghafar-Zadeh et al. [28] used a charge-based capacitance measurement (CBCM) method in a CMOS capacitive biosensor to monitor the growth of *E. coli* in the Luria–Bertani (LB) medium. This chip was integrated with two microfluidic channels implemented by the direct-write fabrication process (DWFP) technique atop on-chip IDEs (as shown in Figure 9c). They achieved a sensitivity of 255 mV/fF, and an LoD of 10^7^ CFU/mL. In another effort, the same group [92] used bacteriophage as a BRE to detect Salmonella and *E. coli*. Another bacteriophage-based bacteria (*E. coli*) detection was reported by Yao et al. [93] using the conductometric method. The sample resistance was converted to frequency and the pulse-width of the output signal was controlled by a one-shot circuit.

#### 4.2.3. Other Electrochemical Sensors

The amperometric and voltammetric methods are two other types of electrochemical sensors that measure the current generated during the reactions between the electrode and the analyte at a constant DC voltage and a variable voltage, respectively. Niitsu et al. [94] presented a high-speed amperometry circuit and two types of electroless plated microelectrode arrays (MEAs) for direct bacteria and HeLa cell counting and even for smaller subjects such as viruses. The MEAs with smaller sizes were expected to be useful even for smaller subjects such as viruses (20 to 970 nm). For noise reduction, the same group [95] implemented a current integrator in conjunction with these bacterial-sized MEAs and could achieve a high SNR of 30.4 dB.

Sun et al. [85] reported a 4096-pixel electrochemical biosensor with an array of gold electrodes fabricated by a 0.18 µm CMOS process, including IDEs (45 µm × 45 µm) surrounded by nano-wells (~9 pL) and a little-used technique called coulostatic discharge sensing for detecting anti-Mumps and anti-Rubella antibodies in human serum. Figure 9d depicts the photograph of this high-density array IC. By opening the passivation across the entire IDE, three 3D trenches were formed between two electrodes, which helped to increase collection efficiency and amplify the signal. As seen in Table 4, they could achieve an LoD of 100 nM.

ISFETs, the most common type of potentiometric sensors, can also be fabricated in standard CMOS technology. For example, as seen in Table 4, Nikkhoo et al. [87] used ISFETs with a post-processed polyvinylchloride (PVC)-based potassium (K^+^)-sensitive membrane as well as two bacteriophages as BREs for the detection of live bacteria (*E. coli*) at two different temperatures in less than 10 min. They used a source and drain follower for the readout circuit.

### 4.3. Magnetic Sensor

Magnetism originating from the target of interest labelled with magnetic particles can be detected from the samples which are free of magnetic background. In these sensors, as seen in Figure 10a, samples and transducers are not in direct contact because the magnetic field can penetrate the insulating layers. As a result, the hardware preparation of the CMOS chip should be simplified before the assay [82].

Various transducers have been reported to convert the sensed magnetism to an electrical signal. Hall sensors are useful for sensing the magnetic field and converting it to current or voltage signals. The hall sensor made of n-type silicon can be fully CMOS-compatible and provide moderate mobility. The Hall effect is the deflection of the current carriers in a semiconductor with the current flowing orthogonal to a magnetic field. The magnetic field applies a Lorentz force to the current carriers, resulting in charge deflection, which allows for electronic detection [82]. For example, Aytur et al. [30] proposed a CMOS-based magnetic bead bioassay platform consisting of a 1024-element array of Hall sensors for infectious disease diagnosis by an immunological recognition similar to ELISA. For the clarity of this biosensor, an aluminum layer was omitted and a gold overlay was deposited on its surface. The biosensor was placed into the gap of a custom electromagnet core which can be operated in either a DC washing mode for removing specifically bound beads from the sensor surface or an AC measurement mode for producing local magnetic fields detectable by the sensor. Figure 10b compares the sensor immunomagnetic bead assay and ELISA. As seen in Figure 10b and Table 4, experimental results demonstrated that this platform was capable of capturing antigen of purified mouse IgG and detecting human anti-dengue virus IgG.

In another work, Pai and Wang et al. [31,96] used an embedded LC-oscillator in a frequency-shift-based CMOS magnetic biosensor with silicon nitride [31] and PDMS [96] surfaces for antigen and DNA detection (see Figure 10c,d). They employed this sensor for amplification-free detection of the interferon-γ protein, which is relevant for tuberculosis diagnostics as well as for a DNA oligonucleotide. The inductor of the circuit shown in Figure 10c was implemented on the top metal layer of the CMOS chip. Probe-target binding changes the effective inductance of the inductor, which subsequently changes the measured frequency. They also proposed a magnetic freezing technique to neutralize the effect of magnetic beads on the sensor and improve SNR. As seen in Table 4, the circuit which was fabricated by the 0.13 µm CMOS process occupies 2.95 µm × 2.56 µm and consumes a total power of 165 mW.

To recapitulate, each type of CMOS biosensor can be selected based on the requirements and the desired approaches, which have been discussed in the previous sections. Moreover, in some cases, multiple sensors might be useful to achieve complementary results for more accurate analyses. For these cases, CMOS technology makes it possible to integrate various types of sensors on a single chip. CMOS technology also paves the way for rapid and high-throughput measurements.

Generally speaking, both optical and magnetic biosensors require optical and magnetic labels which make the biosensing process more sophisticated and expensive. Furthermore, most of the optical biosensors are still bulky in comparison to electrochemical sensors. However, smartphone-based techniques that can target antigen or antibody can provide excessive surveillance and communication and have great potential as home-used PoC testing. Although most protein-based assays for SARS-CoV-2 detection take advantage of optical sensors, the merits of electrochemical biosensors, especially FET-based biosensors, including being label-free and scalability, have drawn the attention of many researchers for the purpose of developing electrochemical immunosensors functionalized with virus antibodies or antigens for the detection of respiratory viral infections (as mentioned in Table 2 and Table 3).

Table 4 shows that Si, SiO_2_ and gold are among the most favorite materials fabricated above the CMOS biosensor chips as the transducer surface material and, according to Table 2 and Table 3, they can also be used for the functionalization of antibodies or antigens for respiratory viral infections such as COVID-19. However, the fabrication of gold electrodes requires post-CMOS processing steps, while Si, SiO_2_ and Si_3_N_4_ can be fabricated in the CMOS process without post-CMOS processing steps. Developing more efficient surface materials adaptable to the CMOS fabrication process is a vital issue for future research.

For the platforms intended only for research, post-processing facilities or specialized microfluidic packaging techniques [8,97] have usually been used to modify CMOS and provide a biocompatible package. However, to enable the adoption of CMOS as the basis of commercial PoC devices and biosensors, only low-cost, post-processing techniques must be used. Moreover, affordable and generic microfluidic packaging methods are required to be incorporated with reusable CMOS biosensors.

The CMOS readout circuits reviewed in this section can be reconfigured and calibrated based on the required design metrics required for the specific application. However, they should be further improved in terms of reproducibility, reusability, low LoD, high throughput, and fast response, making them more trustworthy for mass production.

Optimization of the power consumption and incorporating portable power sources would also advance their functionality, especially for the areas that are far from power sources. There is also a growing interest in developing wearable sensors with flexible substrate materials and wireless communication capability which enables mass use and reports through the Internet of Things (IoT).

To the best of our knowledge, there is no commercialized CMOS-based PoC device for COVID-19 applications. However, the related research works reviewed in this paper show that recent advances in CMOS-based biosensors have opened a new avenue for the development of portable low-cost CMOS-based PoC platforms in urgent pandemic situations.

## 5. Conclusions

COVID-19, as a major health threat worldwide, has raised important questions for researchers about how they could develop a globally affordable platform for the timely and precise diagnosis of these diseases. Cutting-edge technologies such as CMOS biosensors can open a new avenue to achieve this goal. In this review, we tried to study the recent advances in protein-based test kits dedicated to COVID-19, as well as the diverse types of CMOS biosensors which are potentially useful for detecting dangerous pathogens.

Immunosensors, by offering the advantages of direct antigen detection, good specificity, and sensitivity, can be the alternatives for PoC devices. However, their limitations, such as cross-reactivity and the like, should be avoided. These sensors should be able to differentiate between coronavirus variants and the agents of other seasonal respiratory diseases.

A CMOS biosensor including the transducers and the readout circuits on a single chip, which is incorporated in a microfluidic platform, can provide short time workflow, high-throughput measurement, low LoD, high resolution, low power consumption, portability, automation, multiplexing, and the parallel detection of a series of parameters. Although there is a considerable amount of literature on CMOS-based biosensors as a promising candidate for the development of PoC devices, further experimental investigations are still required to overcome the bottleneck of standard diagnostic tools and achieve a generic and low-cost hybrid CMOS-microfluidic platform with a swift response, superb reliability and wide availability that can be quickly translated to new strains or novel viruses during a pandemic outbreak. Furthermore, they should be reusable and user-friendly devices for the general community to be able to carry out self-testing and communicate with healthcare centers helping them to self-isolate and quarantine themselves. Therefore, in the future, more studies should concentrate on simplifying all user steps at a minimal cost.

## Figures and Tables

**Figure 1 micromachines-12-00915-f001:**
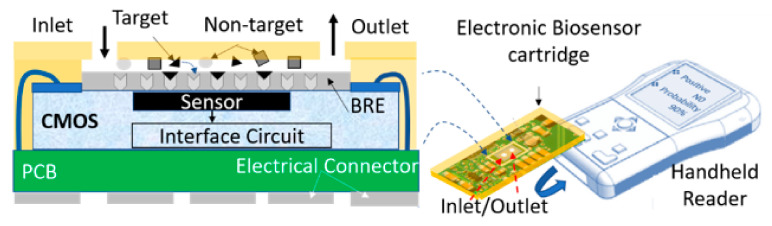
Schematic view of fully integrated CMOS-based PoC system including an electronic disposable cartridge and a reader: CMOS biosensor features a CMOS sensor and circuit, BRE layer, and microfluidic with inlet/outlet.

**Figure 2 micromachines-12-00915-f002:**
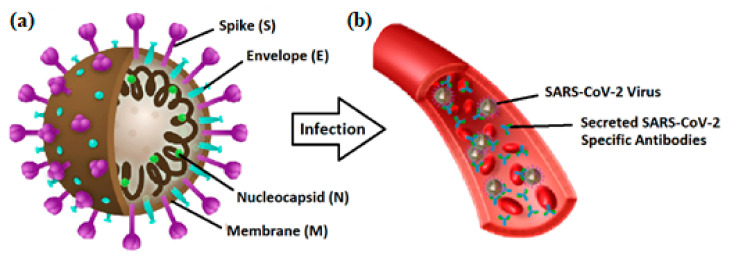
Schematic view of (**a**) SARS-CoV-2 virus and its structural proteins; (**b**) infection by SARS-CoV-2 and secretion of specific antibodies in the bloodstream.

**Figure 3 micromachines-12-00915-f003:**
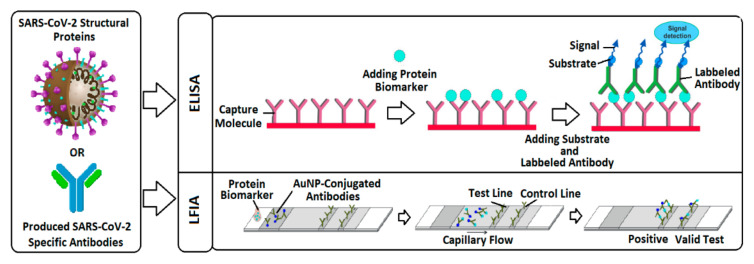
Schematic view of two main protein-based detection kits: ELISA employs capture molecules attached to the surface of the plates and detects the protein biomarker using specific labelled antibodies. Detection takes place by electronic plate reader, which is more accurate, or by the naked eyes. LFIA consists of a nitrocellulose strip which includes specific capture molecules in test line and control line and employs AuNPs for visual detection.

**Figure 4 micromachines-12-00915-f004:**
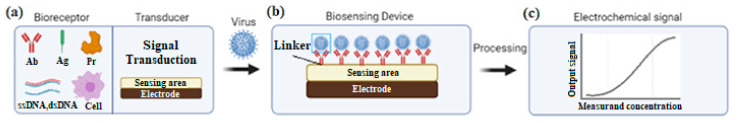
Schematic view of an electrochemical biosensor: (**a**) The surface of these biosensors can be functionalized with various receptors; (**b**) when spotted on the sensing area, these BREs capture the specific targets such as viral particles; and (**c**) the electrochemical signal is reported after the processing steps. Ag: Antigen, Ab: Antibody, Pr: Protein.

**Figure 5 micromachines-12-00915-f005:**
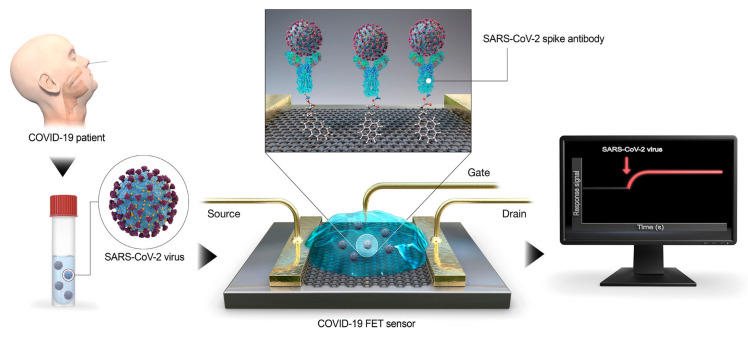
The developed G-FET biosensor for COVID-19 detection. The surface is coated with graphene as the sensing area and 1-pyrenebutyric acid N-hydroxysuccinimide ester is used as the probe linker to functionalize the graphene surface with SARS-CoV-2 anti-S antibody [59].

**Figure 6 micromachines-12-00915-f006:**
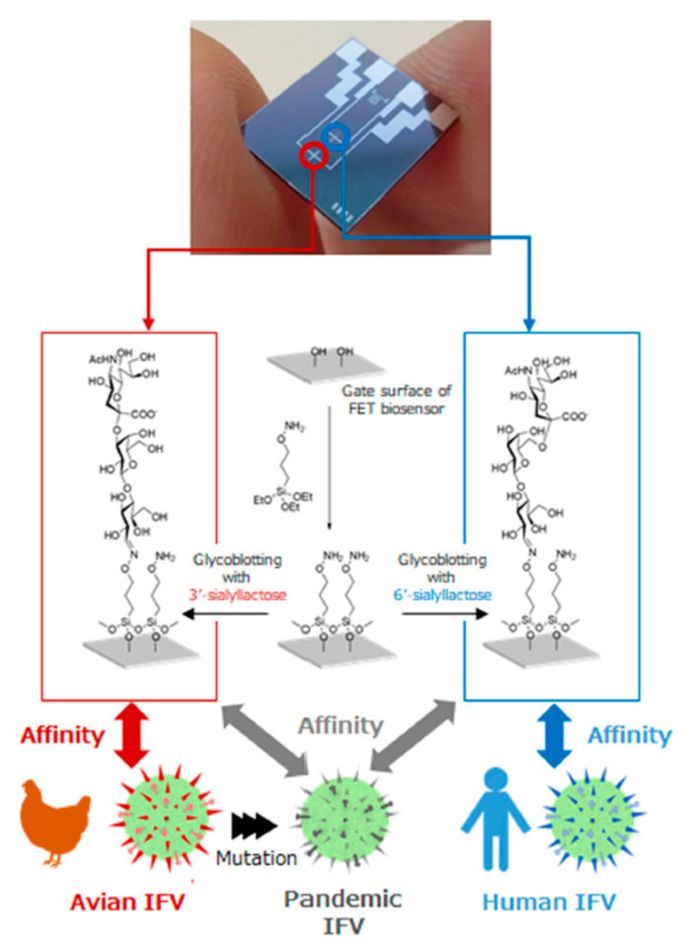
Schematic view of the developed dual-channel field-effect transistor for detection of the Influenza virus and the steps of surface functionalization for specific targets [75].

**Figure 7 micromachines-12-00915-f007:**
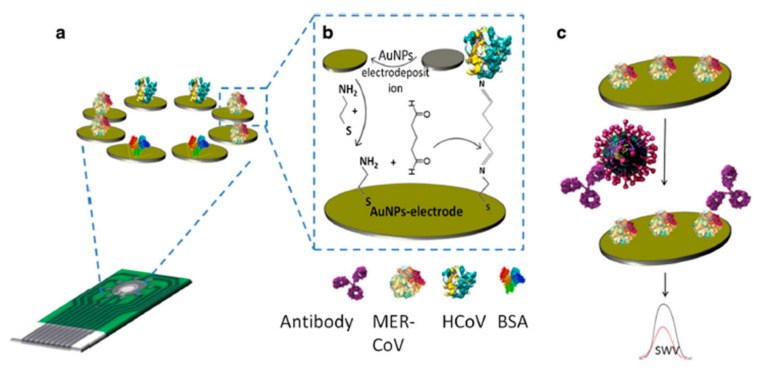
(**a**) An immunosensor for MERS-CoV detection; (**b**) fabrication of the biosensor and its surface functionalization; (**c**) virus detection process [79].

**Figure 8 micromachines-12-00915-f008:**
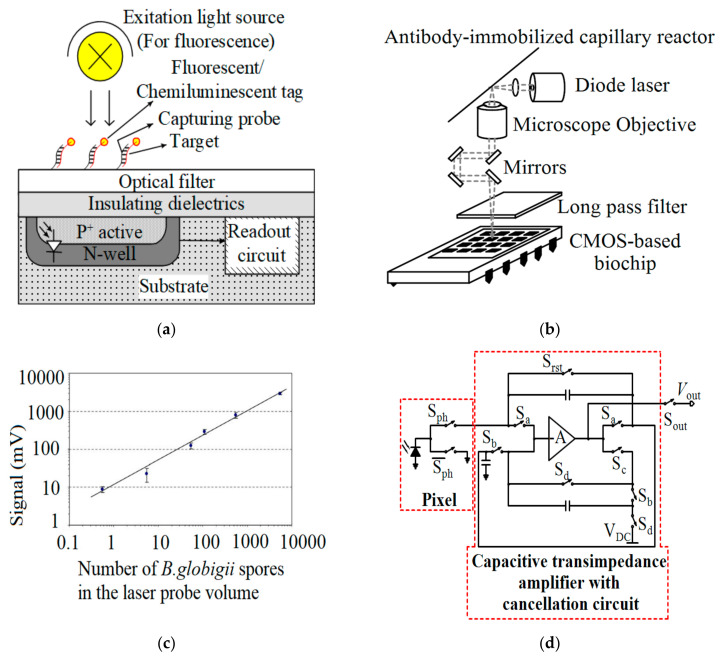
(**a**) CMOS-based optical sensor; (**b**) The diagram of the biochip proposed by Song et al.; (**c**) Detection of *B. globigii* in an antibody-immobilized capillary reactor utilizing the system shown in (**b**) [35]; (**d**) Schematic of the pixel and processing architecture of the LFIA reader proposed by Pilavaki et al.

**Figure 9 micromachines-12-00915-f009:**
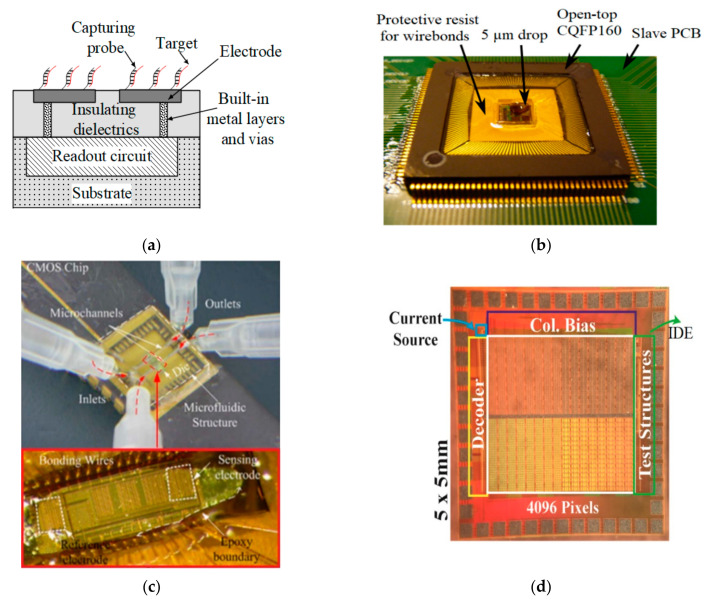
(**a**) Interfacing techniques for electrochemical sensing; (**b**) Packaged CFC chip proposed in [91]; (**c**) A capacitive sensor chip packaged by DWFP technique proposed in [28]; (**d**) The photograph of the 64 × 64 redox amplified coulostatic discharge-based biosensor array reported in [85].

**Figure 10 micromachines-12-00915-f010:**
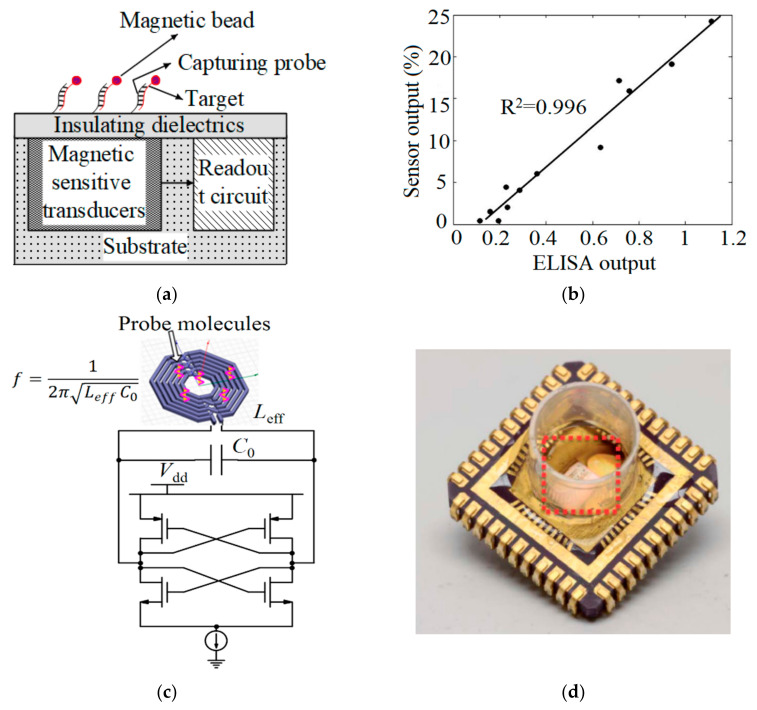
(**a**) CMOS-based magnetic biosensor; (**b**) Comparison of anti-dengue virus IgG detection between ELISA and the sensor immunomagnetic bead assay proposed in [30]; (**c**) Frequency-shift magnetic sensor; (**d**) The disposable cartridge including an electrically connected magnetic-based biosensor chip inside a polypropylene well [31].

**Table 1 micromachines-12-00915-t001:** FDA EUA-approved protein-based assays for SARS-CoV-2 detection.

Website	Technology	Biomarker (Protein)	Sensitivity	Specificity	PPA	NPA	Time (min)	Technology Highlights
[22]	ELISA- Color change	IgA, IgG, and IgM antibodies	97.5%	99.06%	97.5%	99.1%	>120	*Advantage:* High-throughput Well-stablished technology Detects both current and previous infection *Disadvantage:* Requiring specialized personal time consuming Costly Many manuals’ steps increasing the error risk Laboratory-based
[23]	LFIA- AuNP Antibody detecting cassette	IgM/IgG antibodies	93.8% after day 7	96.0%	93.8%	96.0%	10-20	*Advantage:* PoC test Cost-effective Rapid Useful for disease follow-up Requiring small sample volume Room temperature storage *Disadvantage*: Low accuracy Analyzes one sample per test Not useful for early detection False results specifically when tested in the early phases of the infection In symptomatic cases, negative result requires RT-PCR For confirmation
[24]	LFIA- AuNP Antibody detecting cassette	IgM/IgG antibodies	50% at Day 1~6, 91.7% after Day 7	97.5%	60.0%	98.8%	10-15	
[16]	LFIA- AuNP Antigen detecting cassette	Virus Nucleocapsid antigen	97.1%	98.5%	97.1%	98.ase5%	15	*Advantage:* PoC test Cost-effective Rapid Low sensitivity High specificity Useful for detection in asymptomatic cases and before the symptom initiation *Disadvantage:* False results Not useful for early detection In symptomatic cases, negative result requires RT-PCR as confirmation.
[25]	Immunoassay- Fluorescent detection	Virus Nucleocapsid antigen	100% in three first days, 97.6% on day 12	96.6%	97.6%	96.6%	12	*Advantage:* Small sample volume High accuracy More accurate and cost-effective than ELISA Simultaneous detection of multiple targets Simple designs *Disadvantage:* Laboratory-based Requiring manual steps Requiring an additional reader device

**Table 2 micromachines-12-00915-t002:** The recent immunosensors functionalized with antibodies for the detection of respiratory viral infections.

Virus	Target	Biorecognition Element	Biosensor Type	Surface	Linker	LoD	Specificity	Time	Sample	Year Ref.	Notes
SARS-CoV-2	S protein	Ab against SARS-CoV-2 S protein	Electrochemical—G-FET	Si/SiO_2_/Graphene	PBASE	242 particles/mL	No measurable cross-reaction (with MERSCoV Antigen)	-	Nasopharyngeal swabs, no sample preparation required	2020 [59]	*Electrochemical sensors:* Useful for rapid PoC detection Highly sensitive Low LoD Instantaneous measurement Requiring low amount of analyte G-FET sensors are beneficial for ultrasensitive and low-noise detection SWCNT offers high on-state conductance and on/off ratio, and demonstrates higher sensitivity compared with other carbon nanomaterials MOSFET accepts saliva samples and has achieved a measurement time of 10 ms *Optical sensors:* Highly sensitive specially for larger target molecules such as antibodies Quantitative analysis P-FAB is a technology based on U-bent optical fiber sensor, useful for developing rapid and low cost PoC diagnostic tools. P-FAB can be developed in both label-free and labelled formats. LSPCF is a combination of LSP and sandwich immunoassay, promising for early detection and capturing very low concentrations of the analyte
	S protein (S1 subunit)	Ab against SARS-COV S protein (S1 subunit) (CSAb)—COVID-19 S protein (S1 subunit) Ag	Electrochemical—G-FET	Graphene	No info.	0.2 pM	No measurable cross-reaction	~2 min	S1 solution in PBS	2020 [68]	
	SAg and NAg	SARS/SARS-CoV-2 S protein (sub-unit 1) polyclonal Ab and anti-N protein Ab	Electrochemical SWCNT—FET	Si/SiO_2_	EDC/sulfo-NHS	0.55 fg/mL for SAg and 0.016 fg/mL for NAg	Minimal responses to nonspecific proteins	<5 min	Nasopharyngeal swabs, no sample preparation required	2021 [69]	
	S protein (S1 subunit)	SARS-CoV-2 Ab	Electrochemical—MOSFET	Gold-plated carbon electrodes	TGA functionalized electrode was submerged in N,N0-dicyclohexylcarbodi-imide and N-hydroxysuccinimide	100 PFU/mL	-	15 min	Two different purchased antibodies	2021 [70]	
	N protein	anti-N protein mAb	Optical—P-FAB	U-bent fiber-optic Probe (silica fiber)	Thiol-PEG-NHS	10^6^ particles/mL	Label-free biosensor: poor specificity Labelled biosensor: best possible specificity	5 min for labella and 15 min for label-free bioassay	Patient’s saliva sample, requiring minimal preparation process	2020 [71]	
	S protein	nCovid-19 mAb	Electrochemical—eCovSens (PCB-based)	Glass surface coated with fluorine doped tin oxide	Immobilized homogenous layer of AuNPs	90 fM	No cross reactivity with HIV, JEV, and AIV antigens	10-30 s	Spiked saliva samples	2020 [72]	
SARS-CoV	N protein	AMP (Fibronectin)	Electrochemical—FET	Si/SiO_2_/In_2_O_3_NWs	EDC	100 nM	-	10-15 min	N solution in PBS	2009 [73]	
	N protein	Anti-SARS-CoV N-1 mAb	Optical—LSPCF	PMMA optical fiber	Ethyl acetate	1.00 pg.mL^−1^	Higher than other immunoassays such as single capture and labelinh	-	Recombinant SARS-CoV N protein in PBS buffer	2009 [74]	
Influenza A virus	Human H1N1 and avian H5N1 IFV particles	6′-sialyllactose and 3′-sialyllactose	Electrochemical—Dual-channel FET	SiO_2_	Sialic acid-α2,6-galactose and sialic acid-α2,3-galactose	10^0.5^ TCID_50_/mL	Detects Newcastle disease virus (NDV) as well	-	Mucus samples, preparation includes mixing the nasal mucus with virus suspension	2019 [75]	
	Virus particle	mAb of the H1N1 virus	Electrochemical—Nanonet FET	SiO_2_	Anhydrous ethanol with APTES/glutaraldehyde	10 pg/mL	Negligible non-specific bindings	20 min	H1N1 virus solutions in PBS	2019 [76]	
	Virus particle	mAb against H5N2 virus	Electrochemical—SiNW-FET	SiO_2_/Si	MPTMS	~3 × 10^4^ particles/mL	No cross-reaction	40 min	AIV solution in PBS	2012 [77]	

SARS-CoV-2: Severe acute respiratory syndrome coronavirus 2, SARS-CoV: Severe acute respiratory syndrome coronavirus, S protein: Spike protein, N protein: Nucleocapsid protein, G-FER: Graphene field effective transistor, PBASE: 1-pyrenebutanoic acid succinimidyl ester, Ab: Antibody, Ag: Antigen, SAg: S antigen, NAg: N antigen, SWCNT: Single-walled carbon nanotube, EDC: N-ethyl-N′-dimethyl aminopropyl carbodiimide, sulfo-NHS: N-hydroxysulfosuccinimide, MOSFET: Metal–oxide–semiconductor field-effect transistor, P-FAB: Plasmonic fiber-optic absorbance biosensor, AMP: Ab mimic proteins, LSP: Localized surface plasmon, LSPCF: localized surface plasmon coupled fluorescence, PMMA: Polymethyl Methacrylate, IFV: Influenza virus, mAb: Monoclonal antibody, SiNW: Silicon nanowire, MPTMS: 3-mercaptopropyltrimethoxysilane, PBS: Phosphate-buffered saline, AIV: avian influenza virus.

**Table 3 micromachines-12-00915-t003:** The recent immunosensors functionalized with virus antigens for the detection of respiratory viral infections.

Virus	Target	Biorecognition Element	Biosensor Type	Surface	Linker	LoD	Specificity	Time	Sample	Year Ref.	Notes
SARS-CoV-2	Anti-SARS-CoV-2 Ab	SARS-CoV-2 recombinant N protein	Optical—SPR	Gold surface	EDC/NHS (surface modified with a monolayer of 3-mercaptopropionic-Leu-His-Asp-Leu-His-Asp-COOH)	~1 μg/mL	-	15 min	N protein solution in PBS	2020 [78]	Portable Device Label-free Rapid—15-min duration The highest response for antibody detection: 226 RU Increase in rN protein concentration on the surface: decrease in antibody detection Steric hindrance has decreased access to rN binding site in higher concentrations
MERS-CoV	Ab for MERS-CoV	S protein (S1 subunit)	Electrochemical—SWV	AuNPs deposited on carbon array	Cysteamine/glutaraldehyde	0.4 pg.mL^−1^	No cross reaction	20 min	MERS-CoV antigen solution in PBS	2019 [79]	Antibody binding to the BREs: reduces the SWV reduction peak current and consequently decreases the current. No response was observed for control electrodes Non-significant adsorption was not detected on the sensors The sensor demonstrated good repeatability and stability after 14 days
HCoV	Ab for HCoV	HumanCoV proteins	Electrochemical—SWV	AuNPs deposited on carbon array	Cysteamine/glutaraldehyde	1 pg.mL^−1^	No cross reaction	20 min	HCoV antigen solution in PBS		

Ab: Antibody, SPR: Surface plasmon resonance, EDC: N-ethyl-N′-dimethyl aminopropyl carbodiimide, NHS: N-hydroxysuccinimide, SWV: Squarewave voltammetry, SARS-CoV-2: Severe acute respiratory syndrome coronavirus 2, SARS-CoV: Severe acute respiratory syndrome coronavirus, HCoV: Human coronavirus, MERS_CoV: Middle East respiratory syndrome coronavirus.

**Table 4 micromachines-12-00915-t004:** CMOS-based biosensors reported for the diagnosis of various infectious diseases.

Application	Detection Target	Technique	Sensor Surface	CMOS Tech.	Area	Array/Pixel	Power (Vdd)	Some Other Features	Ref.
Diagnosis of infectious disease (Dengue)	Antigen of purified mouse IgG and human anti-dengue virus IgG	Magnetic (Hall sensor)	Gold	0.25 µm	2.5 mm × 2.5 mm	1024	-	AT = 30 s for 120 pixels	[30]
Rubella and mumps virus detection	Capsid protein	Electrochemical (Coulostatic discharge sensing)	Gold	0.18 µm	5 mm × 5 mm	64 × 64	95 mW (2.5 V)	LoD = 100 nM	[85]
The reader of LFIA for PoC diagnostics of Influenza A nucleoprotein	Influenza A nucleoproteins	Optical (LFIA reader)	-	0.35 µm	12.28 mm^2^ *	4 × 64	21 µW (2 V)	RN = 1.9 mVrms SNR = 50 dB, FR = 67 fps	[33]
Detection of a bacterial virus	M13KO7	Electrochemical (capacitive)	Si/SiO_2_	1.5 µm	-	1	-	-	[86]
Detection of single bacterial cell	*S. epidermidis*	Electrochemical (Capacitive)	Al_2_O_3_	0.25 µm	14 µm × 16 µm	16 × 16	29 µW (2.5 V)	SNR = 37 dB, LoD ~ 7 bacteria (450 aF), Sensitivity = 55 mV/fF (2.2 mV/bacteria), IDR = 0.45 fF to 57 fF	[26]
Detection of *B. globigii* spores based on the combined use of ELISA and LIF detection	*B. globigii* spores	Optical (LIF)	Silica capillaries	-	-	4 × 4	-	LoD = 0.55 cells/probe	[35]
Detection of *S. pneumonia* by the measurement of IgG antibody concentrations in human blood sera	IgG antibody	Optical (Chemiluminescence/fluorescence imaging)	SiO_2_	0.5 µm	-	4 × 8	-	-	[34]
Detection of *E. coli*	*E. coli*	Electrochemical (K+-sensitive FET)	SiO_2_	0.18 µm	1.5 mm × 0.6 mm	6	-	AT < 30 Min	[87]
Tuberculosis diagnostics	Interferon-γ protein	Magnetic (frequency-shift based sensing)	Silicon nitride	0.13 µm	2.95 µm × 2.56 µm	8	165 mW	LoD = 1 pM	[31]

* Overall pixel array area, RN = Read noise, FR = Frame rate, IDR = input dynamic range, AT = analysis time.

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
