# Peer review of "Towards Fully Integrated Portable Sensing Devices for COVID-19 and Future Global Hazards: Recent Advances, Challenges, and Prospects"

_micromachines, 2021, doi:10.3390/mi12080915_

Round 1
Reviewer 1 Report
The manuscript entitled: “Towards Fully Integrated Portable Sensing Devices for 2 COVID-19 and Future Global Hazards: Recent Advances, Challenges, and Prospects” refers to the engaged research progress about COVID-19 sensing devices during the last year. The subject area of the manuscript is interesting, and it would certainly add a scientific contribution to the relevant field. I recommend the publication of this manuscript after revision. The following comments/suggestions are provided for the authors' revising manuscript.
- A number of review articles have been recently published on the topic and in this reference, authors are suggested to compare what are lacking in earlier reports that necessitated the present review articles. This should be compared in the introduction sections.
- Quality of Figure 1 should be improved.
- The authors are recommended to read and cite closely related articles to improve this manuscript. More recent literatures should be included to support this review such as; Chemical Engineering Journal, 414 (2021) 128759; Current Pharmaceutical Design, 27 (2021) 1170-1184; Current Opinion in Electrochemistry, 2020, 23: 174–184; Current Research in Chemical Biology, Volume 1, (2021), 100001; Chemical Engineering Journal, 423 (2021) 130189; Current Opinion in Colloid & Interface Science, 2021, 52: 101418.
- All the schemes/figures are adopted from published works, but authors have not included their own opinion. Authors also need to cite the proper references for the adopted figures which are ethical.
- Authors should give critical analysis about detection approaches, pros and cons in tabular form.
- The first area is the pathogenesis of COVID-19, how its effects the human body? Better to add one paragraph about it.
- Conclusion and future prospects should be presented separately while focusing numerous challenges. In future perspective, Authors should elaborate challenges being faced in biosensing devices and their possible solution.
- There are some typographical errors in the paper. The manuscript must be rechecked for the better understanding.
Author Response
Dear Reviewer,
Thank you for reviewing the present article. Your comments significantly helped us to improve the previous version of our review paper. We hope you find the final version satisfying and interesting.
Sincerely,
Tina Shaffaf

Reviewer 2 Report
The authors present a manuscript, titled "Towards Fully Integrated Portable Sensing Devices for COVID-19 and Future Global Hazards: Recent Advances, Challenges, and Prospects Authors". The work is highly timely and should be of interest to the community at large. The manuscript is reasonably described. Overall the flow of the manuscript is good. With that I have a number of comments that require authors attention before considering publication.
1. What is the merit of CMOS-based biosensors compared with other types of biosensors? A brief statement on the comparison between them should be included in the introduction. It means that they should clearly show the advantage of CMOS-based biosensors.
2. One of the key factors of the biosensor is selectivity. The authors should do a careful study of the selectivity of the sensors to ensure that they are indeed measuring what they think they are measuring. This is particularly important since the subject studies shown is very limited.
The authors need to show how various other analytes present in nasopharyngeal swab specimen could effect the sensor response, and clearly demonstrate the selectivity of their sensors against relevant parameters and analytes. In other words, the authors should provide a table and/or graph that shows the selectivity test based on previous studies.
3. The authors should add the key metrics (response time, sample preparation time, etc.) at the table 2 and 3. These metrics will further help readers compare sensor performance.
Author Response

(The authors gave the same response as above.)

Reviewer 3 Report
This manuscript reports the recent advancements of CMOS biosensors and their application in COVID-19 detection. The authors reviewed various types of CMOS-based biosensing including electrochemical biosensing, optical biosensing, capacitive biosensing, magnetic biosensing, and their applications in COVID-19 diagnosis.
- It is reasonable that the authors try to claim the review does not only contribute to COVID-19 but can be more general. However, the authors do not describe or explain this logic very clear. The authors have to decide to either only focus on COVID-19 biosensing or rewrite the title, abstract, introduction to make this clear.
- The authors need to redo the references. Figures that were not originally from the authors also should be cited. How to cite other groups’ work also need to be careful.
- Review the work from antigen and antibody separately seems reasonable. However, it may be valuable to add another table to summarize and compare different technologies, such as electrochemical, optical, magnetic, etc. Like Table 3, but need further editing.
- As a review paper, the authors are expected to look beyond the mainstream technologies and discussed the advantages, disadvantages, challenges, and possible paths toward the final solution.
Minor concerns:
- The abstract need some edits. “CMOS technology…” could be shifted to the end that right before the author claim this manuscript gives an overview of the CMOS biosensors.
- Introduction, “despite…”. What are those challenges?
- Introduction, “… fully integrated sensing technologies…” is confused. Commercial products or even bulky equipment in hospitals are also fully integrated sensing technologies. Do you mean “fully integrated CMOS-based PoC system” instead? Before presenting the claim “fully integrated CMOS-based PoC system”, the authors should review or explain the motivation. So the paragraph “CMOS by offering the striking….” Should be presented ahead of the claim or the proposal.
- Introduction, “Section 4 will put forward….”, “put forward” here is not accurate.
- Figures (Fig.4, Fig.8, Fig.9, Fig.10) need further editing, the texts in the figures and figures are not clear. Why only focus on electrochemical methods here?
- “3,1” should be antibody?
- References should also be added to the caption of the figures.
- “With our recent study…” in reference [58] a publication from the authors’ group?
- Section 4 needs further editing. The organization is confusing. “Capacitive sensing” does not belong to electrochemical sensing.
- The last section should be section 5.
- Further analysis that predicts the existing challenges and possible solutions toward the final solution should be added to the conclusion section.
Author Response

(The authors gave the same response as above.)

Reviewer 4 Report
This work by Shaffaf et al describes the recent progress in immunosensors for the diagnosis of covid 19. The main contribution of this work is the very useful tables that go over the different sensors, covering electrochemical and optical sensing techniques. I support the publication of this work after the following minor revisions are addressed.
- Include more clearly in the introduction a sentence clarifying how this review is different from other reviews on this topic.
- Expand the conclusion and outlooks sections. This section should be used to provide critical insight into the challenges and opportunities in the field. what in your opinion is the most likely method to be used in everyday testing?
- A lot of figures don't reference the paper in the caption (eg fig 5) and don't indicate any copyright or permission obtained. please update and be sure to secure all proper permissions, eg(Copyright 2012, American Chemical Society.
- I suggest including a description of Crisper based covid sensing eg (CRISPR–Cas12-based detection of SARS-CoV-2. Nature biotechnology, 38(7), pp.870-874), as this type of technique has been developed as a commercial enterprise and has been used in clinic,
Author Response

(The authors gave the same response as above.)
